# Antimicrobial Effect of *Moringa oleifera* Leaves Extract on Foodborne Pathogens in Ground Beef

**DOI:** 10.3390/foods12040766

**Published:** 2023-02-09

**Authors:** Reda Abdallah, Nader Y. Mostafa, Ghada A. K. Kirrella, Ibrahim Gaballah, Kálmán Imre, Adriana Morar, Viorel Herman, Khalid Ibrahim Sallam, Hend Ali Elshebrawy

**Affiliations:** 1Department of Food Control, Faculty of Veterinary Medicine, Kafrelsheikh University, Kafr El Sheikh 33516, Egypt; 2Department of Animal Production and Veterinary Public Health, Faculty of Veterinary Medicine, Banat’s University of Agricultural Sciences and Veterinary Medicine “King Michael I of Romania” from Timisoara, 300645 Timisoara, Romania; 3Department of Infectious Diseases and Preventive Medicine, Faculty of Veterinary Medicine, Banat’s University of Agricultural Sciences and Veterinary Medicine “King Michael I of Romania” from Timisoara, 300645 Timisoara, Romania; 4Department of Food Hygiene and Control, Faculty of Veterinary Medicine, Mansoura University, Mansoura 35516, Egypt

**Keywords:** *Moringa oleifera*, beef meatballs, sensory attributes, shelf life, foodborne pathogens

## Abstract

Consumers nowadays are becoming more aware of the importance of using only meat products containing safe and natural additives. Hence, using natural food additives for extending the shelf life of meat along with delaying microbial growth has become an urgent issue. Given the increasingly popular view of *Moringa oleifera* leaves as a traditional remedy and also the scarcity of published data concerning its antimicrobial effect against foodborne pathogens in meat and meat products, we designed the present study to investigate the antimicrobial effect of *Moringa oleifera* leaves aqueous extract (0.5%, 1%, and 2%) on ground beef during refrigerated storage at 4 °C for 18 days. MLE revealed potent antimicrobial properties against spoilage bacteria, such as aerobic plate count and Enterobacteriaceae count. MLE 2% showed a significant (*p* < 0.01) reduction in the counts of *E. coli* O157:H7, *Salmonella enterica* serovar Typhimurium, and *Staphylococcus aureus* artificially inoculated to ground beef by 6.54, 5.35, and 5.40 log_10_ CFU/g, respectively, compared to control, by the 18th day of storage. *Moringa* leaves extract (MLE) had no adverse effect on the overall acceptability and other sensory attributes; moreover, it induced a slight improvement in the tenderness and juiciness of treated ground beef, compared to the control. Therefore, MLE can be used as a healthy, natural, and safe preservative to increase meat products’ safety, quality, and shelf stability during cold storage. A promising approach for using natural food additives rather than chemical preservatives could begin new frontiers in the food industry, as they are more safe and do not constitute health risks to consumers.

## 1. Introduction

Meat is a prime source of high-biological value protein, vitamins, and minerals. Despite being nutritious, its high moisture content, water activity, and suitable pH render meat an excellent medium for microbial growth and lipid oxidation, which induces shelf-life quality deterioration [1]. The microbial contamination of meat occurs mainly during processing, by contact with dirty skin, intestinal contents, contaminated facility equipment, such as knives, saws, and grinders, contact with infected food handlers, or exposure to polluted air and water [2]. Proper handling and preservation could reduce the growth of most spoilage and pathogenic bacteria. Total aerobic plate count (APC) and Enterobacteriaceae count (EBC) could be used as good indicators for determining the quality and safety of meat and meat products. APC is considered a gold standard for estimating the overall bacterial populations, and their higher counts of more than 6 log_10_ CFU/g are associated mainly with poor quality and rapid decomposition of meat [3]. EBC is considered a critical indicator in the food industry for evaluating poor hygienic conditions during meat processing and the degree of fecal contamination [4]. Moreover, high EBCs are generally related to the growth of foodborne pathogens originating from feces [2].

Fresh meat and meat products are likely to experience microbial growth, which has a deleterious effect on nutritional quality, results in enormous economic losses, and is accompanied by unfavorable effects on the meat quality, including offensive odor, color, and changes in the texture of meat products [5,6]. Contaminations of meat by foodborne pathogens, such as *Escherichia coli* O157:H7, *Salmonella* spp., and *Staphylococcus aureus*, represent a potential threat to public health [7]. For example, *Salmonella* spp., *E. coli* O157:H7, and *S. aureus* are the cause of 13.3%, 3.6%, and 2.6% of foodborne illnesses annually in the United States, respectively [8], representing a heavily socioeconomic burden on healthcare systems.

A promising approach to using natural food additives, such as organic acids (e.g., sorbic, propionic, citric acid), bacteriocins from microbial sources (nisin, natamycin), enzymes obtained from animal sources (e.g., lysozyme, lactoferrin), plant extracts and essential oils derived from herbs and spices (e.g., basil, thyme, oregano, cinnamon, sage, clove, lemongrass, marjoram, and rosemary), and naturally occurring polymers (chitosan) have become very popular in the food industry to prolong the shelf-life of meat and meat products and preventing the nutritional and sensory losses induced by microbiological or chemical changes [5]. Nowadays, safe natural additives are preferable to the synthetic chemical additives that may be resulting in health risks to consumers [9]. Therefore, the meat industry pays great attention to a wide and renewable variety of natural antioxidants from plant extracts to improve the shelf stability of meat, as well as delay microbial growth and lipid oxidation [3,10]. Plant extracts have massive bioactive compounds that can damage the cell wall and cytoplasmic membrane of spoilage microorganisms and enhance the physical, chemical, textural, and organoleptic properties of processed meat products [11]. The demand for natural antimicrobials and antioxidants is increasing for meat and meat products. Since ancient times, plant extracts have been used for medical, pharmaceutical, phytotherapy, and sanitary purposes, as well as for the food and beverage industry. Moreover, plant extracts and EOs are considered natural preservatives with strong antioxidant, antifungal, and antimicrobial activities in the food industry for the preservation of raw and processed food [9].

*Moringa oleifera* is a fast-growing, drought-resistant tree of the family Moringaceae, commonly known as drumstick tree (from the long, slender, triangular seed pods) or horseradish tree (from the taste of the roots, which resembles horseradish) [12]. *Moringa oleifera* is an important medicinal herb traditionally used as a vegetable and it is native to India, Africa, Arabia, Southeast Asia, and South America. *Moringa oleifera* has been used since ancient times in diets, due to its several biological properties as an anti-inflammatory, antitumor, anticancer, anti-diabetes, and antimicrobial agent [11]. Furthermore, *Moringa oleifera* is a major source of essential amino acids, vitamin C, tocopherol, beta carotene, and minerals. *Moringa oleifera* provides 14 times the calcium in milk, 9 times the iron in spinach, 7 times the vitamin C in oranges, 4 times the potassium in bananas, and 2 times the vitamin A in carrots [12]. In addition to its nutritional quality, it gives a favorable taste and aroma to foods, and additionally, the presence of phytochemicals, including flavonoids and other phenolics in their leaves extract, can hinder the growth of pathogenic microorganisms and extend the shelf life of food [11].

Owing to the great belief in *Moringa oleifera* for centuries as a miracle tree and traditional remedy for many diseases, and also to the scarcity of published studies about the efficacy of *M. oleifera* extract as a natural food preservative possessing an antimicrobial effect, particularly against foodborne pathogens, the current study aimed to investigate the antimicrobial effect of various concentrations of *Moringa oleifera* leaves extract (MLE) against foodborne pathogens (*Escherichia coli* O157:H7, *Salmonella enterica* serovar Typhimurium, and *Staphylococcus aureus*), along with the determination of the shelf life extension, through the enumeration of aerobic plate count and Enterobacteriaceae count, as well as sensory attributes of ground beef during cold-storage at 4 °C for 18 days.

## 2. Materials and Methods

### 2.1. Collection and Preparation of MLE

Fresh *M. oleifera* leaves were obtained from a local herbal store in Tanta city, El-Gharbia Governorate, Egypt. MLE was prepared according to the technique mentioned by Shah et al. [13]. The leaves were washed well with water to remove any adhering dirt, then dried in a hot air oven at 60 °C and ground into a fine powder in a heavy-duty grinder. The powder was passed through sieve No. 60 and extracted by soaking 400 g of dried powder in 2 L of boiled water at room temperature for 1 h, with frequent stirring with a glass rod. The obtained aqueous extract of *M. oleifera* leaves was filtered by Whatman No. 1 filter paper, and the residue was re-extracted again with 1 L distilled water. Both filtrates were mixed and freeze-dried. The resultant extract was kept in a sterile glass container and stored at 4 °C until use.

### 2.2. Ground Beef Treatment and Preparation for Shelf-Life Determination and Sensory Attributes

Twelve kilograms of fresh beef cuts were bought from a butcher shop in Tanta city, El-Gharbia Governorate, Egypt. Beef cuts were packed in sterile plastic bags, and transferred in an ice box to the Food Microbiology Laboratory, Department of Food Hygiene and Control, Faculty of Veterinary Medicine, Benha University. The fresh beef was cut by a sterile sharp knife and minced using a clean sterile 0.32 cm grinder plate and divided into 4 groups (3 kg each). Three groups were treated with different concentrations of *M. oleifera* leaves aqueous extracts (0.5%, 1%, and 2%) and the fourth group served as control (without any treatment). The treated groups with different concentrations of MLE were mixed thoroughly for five minutes by hand, to have a homogenous mixture. The meat of both control and treated groups was formed into meatballs (25 g each), packaged into sterile impermeable plastic pouches, and kept in the refrigerator at 4 °C for 18 days to examine its sensory attributes and microbial count (aerobic plate count and Enterobacteriaceae counts) to determine its the shelf-life.

### 2.3. Aerobic Plate Counts and Enterobacteriaceae Counts for Shelf Life Determination

Aerobic plate counts and Enterobacteriaceae counts were tested from Day 0 and every 3 days thereafter for a period of 18 days. Ten grams of the control and treated groups were taken and homogenized with 90 mL of sterile 0.1% peptone water (CM0009; Oxoid Ltd., Basingstoke, UK) using a laboratory blender for 2 min. Ten-fold serial dilutions were prepared according to the technique recommended by ISO [14]. Appropriate dilutions were plated on Plate Count Agar (PCA, CM0325; Oxoid Ltd., Basingstoke, UK), and incubated at 30 °C for 48 h to enumerate aerobic plate counts (APC) [15]. Enterobacteriaceae counts (EBC) were enumerated by pouring 1 mL of the appropriate dilutions into a sterile Petri dish containing 15 mL of the violet red bile glucose (VRBGA, CM 0485; Oxoid Ltd., Basingstoke, UK), which had been previously prepared then cooled to 45 °C in a water bath and carefully mix the medium to cool. Once the agar was solidified, an additional 10 mL of medium was added onto the surface of the inoculated plate and incubated at 37 °C for 24 h [16]. The bacterial colonies in the different countable plates of APC and EBCs were counted per gram of all tested samples.

### 2.4. Effect of MLE on Foodborne Pathogens

#### 2.4.1. Bacterial Strains Used and Preparation of Inocula

Three different wild-type foodborne pathogens species previously isolated from meat products in our laboratory were used in this study: *Escherichia coli* O157:H7, *Salmonella enterica* serovar Typhimurium, and *Staphylococcus aureus*. The culture of each pathogen strain was grown in tryptone soy broth (CM0129; Oxoid Ltd., Basingstoke, UK) and incubated for 24 h at 37 °C. One mL of the original dilution of each fresh bacterial culture was transferred in a tube containing 9 mL of sterile 0.1% peptone water (CM0009; Oxoid Ltd., Basingstoke, UK) in a successive manner to make 10-fold dilution series to achieve the desired inoculation level of each pathogenic strain. The appropriate inoculum level of 10^8^ CFU/mL prepared from each foodborne pathogen strain was streaked on the surface of Sorbitol MacConkey Agar supplemented with cefixime and potassium tellurite (CM0813, SR0172; Oxoid Ltd., Basingstoke, UK), Xylose-Lysine-Desoxycholate agar (XLD Agar, CM0469; Oxoid Ltd., Basingstoke, UK), and Baird-Parker selective agar with egg-yolk tellurite emulsion (CM275, S00R54; Oxoid Ltd., Basingstoke, UK) for the isolation of *E. coli* O157:H7, *S. enterica* serovar Typhimurium, and *S. aureus*, respectively. The plates inoculated with foodborne pathogen strains were incubated at 37 °C for 24 h, and specific colonies of the number of the different bacterial pathogens were calculated to detect the inoculation levels, which were found to equal 10.16 ± 0.47, 10.27 ± 0.41, 10.62 ± 0.52, and10.55 ± 0.34 log_10_ CFU/mL for *E. coli* O157:H7, *S. enterica* serovar Typhimurium, and *S. aureus*, respectively. 

#### 2.4.2. Inoculation of Ground Beef Treated with MLE by Foodborne Pathogens

Another 3 kg of fresh beef collected from the same butcher shop were used to detect the antimicrobial effect of MLE on some foodborne pathogens, including *E. coli* O157:H7, *S. enterica* serovar Typhimurium, and *S. aureus*. The 3 kg of purchased ground beef were divided into four groups (750 g each), three of them were treated with different concentrations of MLE (0.5%, 1%, and 2%) and the fourth group served as control (without any treatment). Each of the four groups (three treatments and one control) were subdivided into three subgroups (250 g each) to be inoculated with either *E. coli* O157:H7, *S. enterica* serovar Typhimurium, or *S. aureus*. Each meat group (250 g each) was inoculated with 2.5 mL from each prepared pathogen inoculum (about 10 log_10_ CFU/mL) and thoroughly mixed by hand for 3 min, under aseptic conditions, to gain a homogenous mixture containing about 8 log_10_ CFU/g of each foodborne pathogen. The artificially-inoculated ground beef with the specific foodborne pathogens was formed into meatballs (25 g each), packaged in sterile plastic pouches, and kept in the refrigerator at 4 °C for 18 days. The artificially-inoculated ground beef was examined every 3 days during the storage period (18 days). The isolation and identification of the different foodborne pathogens were conducted according to techniques recommended by ISO [17,18,19] for *E. coli* O157:H7, *S. enterica* serovar Typhimurium, and *S. aureus*, respectively. The counts of samples tested were expressed as CFU/g.

### 2.5. Sensory Evaluation

The sensory assessment of untreated beef meatballs (control group) and treated groups with different concentrations of MLE was conducted on Day 0 (the day of meatballs preparation) and every 3 days thereafter for a period of 18 days. Beef meatballs (25 g each) were put individually in a clean aluminum foil and cooked in an electrical cooking oven at 180 °C for 20 min. The cooked beef meatballs were blind-coded with random numbers and presented to 15 trained panelists from the staff members of the Food Hygiene and Control Department, Faculty of Veterinary Medicine, Benha University, Egypt. An eight-point hedonic scoring scale was used to judge *Moringa* flavor intensity, characteristic flavor for beef meatballs, tenderness, and juiciness (https://www.depts.ttu.edu/meatscience/docs/TTUBeefSensoryForm.pdf) (accessed on 30 June 2022), whereas a nine-point hedonic scale was used to determine the overall acceptability of beef meatballs tested [20].

### 2.6. Statistical Analyses

Triplicate measurement was applied for all samples. The obtained results were analyzed using SPSS software (SPSS Inc., ver. 21, Chicago, IL, USA). The difference between the means of counts of the different microbial categories between various treatments was assessed by one-way analysis of variance. Sensory attribute results were determined using the general linear model (GLM). Results were considered statistically significant at *p*-values < 0.05. 

## 3. Results and Discussion

### 3.1. Antimicrobial Effect of MLE

#### 3.1.1. Effect of MLE on Aerobic Plate Count (APC)

The APC of beef meatballs tested on Day 0 ranged from 3.68 to 3.72 log_10_ CFU/g with no significant difference between the control and treated beef meatballs with 0.5%, 1%, and 2% MLE (Figure 1). However, on Day 3, beef meatballs treated with 0.5%, 1%, and 2% MLE displayed a significant (*p* < 0.05) decrease in APC by 1.16, 1.42, and 1.54 log_10_ CFU/g, respectively, when compared with the control sample. Likewise, on Day 6, treated beef meatballs with 0.5%, 1%, and 2% MLE showed a significant decrease in APC by 1.57, 2.13, and 2.34 log_10_ CFU/g, respectively, compared to the control (4.70, 4.14, and 3.93, respectively vs. 6.27 log_10_ CFU/g) (Figure 1). On Days 9, 12, and 15, treated beef meatballs with 2% MLE revealed a significant (*p* < 0.01) reduction rate of APC by 2.89, 2.92, and 3.27 log_10_ CFU/g, respectively when compared to the control sample (Figure 1). The APC of both control and MLE-treated samples increased with time. Such an increase showed significant differences by Day 3, 9, 12, and 18, and thereafter throughout the storage period for control, 0.5%, 1.0%, and 2% of MLE-treated samples, respectively, when compared with their corresponding initial counts on Day 0. Throughout the storage period (18 days), the APC of all treated beef meatballs with 0.5%, 1%, and 2% MLE remained below 7 log_10_ CFU/g (Figure 1), which is the maximal permissible limit (MPL) for APC in ground beef according to ICMSF [21]. This limit was exceeded in control meatballs on Days 12, 15, and 18 of storage (Figure 1).

At the end of the storage period (Day 18), control meatballs exhibited high APC of 8.15 log_10_ CFU/g, whereas treated beef meatballs with 0.5%, 1%, and 2% MLE showed a significant (*p* < 0.01) decrease in APC when compared to the control sample by 1.60, 2.60, and 3.19 log_10_ CFU/g, respectively (Figure 1). Similarly, Hazra et al. [22] showed that the total plate counts (TPCs) in ground buffalo meat were significantly (*p* < 0.05) decreased after adding MLE at 1.5% and 2%. The same authors mentioned that fresh leaf juice could prevent the growth of microorganisms. Additionally, chicken sausages treated with 0.5%, 0.75% and 1% MLE exhibited significantly (*p* < 0.05) low TPC values throughout the storage period (5 weeks), when compared with chicken sausages treated with 0.25% MLE and control sample [23]. The addition of 1 g/kg *Moringa* leaf extract (ethanolic-aqueous) to ground beef samples kept for 6 days at 4 °C lowered total viable counts (*p* < 0.05) than that in the control and butylated hydroxytoluene (BHT) treated samples by Day 3 of storage [24]. These results indicate that MLE can be used as a natural antimicrobial agent in meat products.

#### 3.1.2. Effect of MLE on Enterobacteriaceae Counts (EBCs)

Enterobacteriaceae are a family of facultatively anaerobic Gram-negative bacteria within the Enterobacterales order that contains many important foodborne pathogens, such as *Salmonella*, *Shigella*, *Escherichia coli*, *Proteus*, *Klebsiella*, and *Yersinia* [25]. The presence of Enterobacteriaceae in meat might be due to poor hygienic conditions during the handling and processing of meat and meat products.

The present study exhibited no significant difference in the initial EBCs (Day 0) between control and treated beef meatballs (Figure 2). However, on Day 3, beef meatballs treated with 0.5%, 1%, and 2% MLE displayed a significant (*p* < 0.05) decrease in EBCs when compared to the control sample by 1.01, 1.23, and 1.47 log_10_ CFU/g, respectively. Similarly, on Day 6, treated beef meatballs with 0.5%, 1%, and 2% MLE showed a significant reduction in EBCs when compared to the control sample by 1.11, 1.36, and 1.54 log_10_ CFU/g, respectively (3.86, 3.61, and 3.43, respectively vs. 4.97 log_10_ CFU/g) (Figure 2). On Days 9, 12, and 15, beef meatballs treated with 2% MLE revealed a significant (*p* < 0.01) reduction rate of EBCs by 1.61, 2.09, and 2.12 log_10_ CFU/g, respectively, when compared with the control sample (Figure 2). The EBC of both control and MLE-treated samples increased with time. Such an increase showed significant differences by Day 3, 12, and 15 and thereafter throughout the storage period for control, 0.5%, and 1.0% of MLE-treated samples, respectively, when compared with their corresponding initial counts on Day 0. Interestingly, the EBCs did not show any significant increase throughout the storage period in 2% MLE-treated samples.

Control meatballs showed high EBCs of 6.17 log_10_ CFU/g on Day 18 (the last day of the storage period), whereas treated beef meatballs with 0.5%, 1%, and 2% MLE exhibited a significant (*p* < 0.05) decrease in EBCs when compared with the control sample by 1.19, 1.76, and 2.17 log_10_ CFU/g, respectively (Figure 2). Similarly, a previous study by Rahman et al. [26] revealed that total coliform count was decreased significantly (*p* < 0.05) amongst goat meat nuggets treated with 0.1%, 0.2%, and 0.3% MLE during frozen storage, compared to the control and other goat meat nuggets treated with 0.1% butylated hydroxyanisole (BHA). Likewise, Mashau et al. [27] observed that at the end of the storage period (Day 15), control mutton patties revealed a high coliform count of 6.20 log_10_ CFU/g, meanwhile, mutton patties treated with 1%, 2%, 3%, and 4% of MLE showed a significant low coliform count of 5.77, 4.88, 3.06, and 2.02 log_10_ CFU/g, respectively, and the same authors found a significant increase in coliform counts in all treated samples, throughout the storage period (15 days). 

### 3.2. Antimicrobial Effect of MLE against Foodborne Pathogens

Studies concerning the antimicrobial effect of MLE against foodborne pathogens are scarce and mostly focused on its in vitro effect against some foodborne pathogens. This study, is therefore, very crucial to elucidate the antibacterial effect of MLE against three of the most important foodborne pathogens in meat products.

#### 3.2.1. Antimicrobial Effect of MLE against *E. coli* O157:H7

There was no significant difference in the initial counts of inoculated *E. coli* O157:H7 in control and treated beef meatballs with MLE, which ranged from 6.59 to 6.69 log_10_ CFU/g. By Day 3, treated beef meatballs with 2% MLE displayed a significant reduction (*p* < 0.05) in *E. coli* O157:H7 counts, by 1.53 log_10_ CFU/g, when compared with the control sample (Table 1). Treated beef meatballs with 0.5%, 1%, and 2% MLE exhibited a significant (*p* < 0.05) reduction in *E. coli* O157:H7 counts on Day 6 of storage, by 1.34, 1.68, and 2.36 log_10_ CFU/g, respectively, compared to the control (6.00, 5.66, and 4.98 respectively versus. 7.34 log_10_ CFU/g) (Table 1). Likewise, on Day 9 of storage *E. coli* O157:H7 counts significantly (*p* < 0.01) decreased by 2.03, 2.80, and 3.70 log_10_ CFU/g, respectively, in meatballs treated with 0.5%, 1%, and 2% MLE when compared to the control (Table 1). By Day 12, beef meatballs treated with 0.5%, 1%, and 2% MLE displayed a significant (*p* < 0.01) decrease in *E. coli* O157:H7 counts by 2.71, 3.39, and 4.5 log_10_ CFU/g, respectively, in comparison with the control sample (Table 1).

By Day 15 of storage, beef meatballs treated with 0.5%, 1%, and 2% MLE revealed a significant (*p* < 0.01) decline in *E. coli* O157:H7 counts by 3.26, 4.27, and 5.39 log_10_ CFU/g, respectively, in comparison with the control (Table 1). On Day 18, control meatballs showed a high count of inoculated *E*. *coli* O157:H7 of 8.54 log_10_ CFU/g, whereas treated beef meatballs with 0.5%, 1%, and 2% MLE exhibited a significant (*p* < 0.01) reduction in *E. coli* O157:H7 counts by 3.70, 4.93, and 6.54 log_10_ CFU/g, respectively, in comparison with the control (Table 1). Interestingly, our study revealed a potent antimicrobial effect of MLE against *E*. *coli* O157:H7 artificially inoculated to beef meatballs. In this context, Moyo et al. [28] found that the *M. oleifera* acetone extract at a concentration of 5 mg/mL had a potent bactericidal effect against multi-drug resistant *E*. *coli* isolates. The in vitro activity of *M. oleifera* leaf extracts showed that MLE had potent antimicrobial activity against Gram-negative bacteria, and the greatest inhibitory effect of the extracts was found towards *E*. *coli* [24]. The low-weight proteins and peptides may be responsible for the antimicrobial activity of *M*. *oleifera* leaves. Furthermore, *E*. *coli* was absent in chicken sausages treated with 0.25%, 0.5%, 0.75%, and 1% *M. oleifera* leaves [23]. Additionally, *E*. *coli* was highest in refrigerated chicken patties without *M*. *oleifera* leaf powder (MLP) compared to chicken patties treated with MLP [29]. Thus, MLE could be used as a potent antimicrobial agent to inhibit *E*. *coli* growth in meat products.

#### 3.2.2. Antimicrobial Effect of MLE against *S. enterica* serovar Typhimurium

There was no significant difference (*p* > 0.05) detected in the counts of inoculated *S. enterica* serovar Typhimurium amongst control and treated beef meatball samples with MLE on Day 0 and Day 3 of the storage period (Table 1). By Day 6, however, beef meatballs treated with 0.5%, 1%, and 2% MLE showed a significant (*p* < 0.05) reduction in *S. enterica* serovar Typhimurium counts in comparison with the control by 1.01, 1.20, and 2.02 log_10_ CFU/g, respectively (Table 1). Treated beef meatballs with 0.5%, 1%, and 2% MLE exhibited a significant (*p* < 0.01) decrease of *S. enterica* serovar Typhimurium counts on Day 9 of storage in comparison with the control by 1.39, 1.85, and 2.73 log_10_ CFU/g, respectively (Table 1). Likewise, on Day 12 of storage, treated meatballs with 0.5%, 1%, and 2% MLE revealed a significant (*p* < 0.01) reduction in *S. enterica* serovar Typhimurium count when compared with the control by 2.07, 2.46, and 3.58 log_10_ CFU/g, respectively (Table 1). By Day 15 of storage, treated beef meatballs with 0.5%, 1%, and 2% MLE showed a significant (*p* < 0.01) reduction in *S. enterica* serovar Typhimurium counts by 2.67, 3.79, and 4.28 log_10_ CFU/g, respectively, compared to the control (Table 1). On the last day of the storage period (Day 18), control meatballs showed a high count of inoculated *S. enterica serovar Typhimurium* of 8.05 log_10_ CFU/g, whereas treated beef meatballs with 0.5%, 1%, and 2% MLE exhibited a significant (*p* < 0.01) reduction in *S. enterica* serovar Typhimurium counts by 3.10, 4.31, and 5.35 log_10_ CFU/g, respectively, when compared with the control (Table 1).

The antimicrobial activity of *M*. *oleifera* may be related to its content of phenolic and flavonoid compounds at concentrations greater than 1 g/kg; additionally, *M*. *oleifera* extract had a potent antimicrobial effect against many foodborne and human pathogens, such as *Salmonella*, *Staphylococcus*, and *Escherichia coli* [24]. Our findings were in line with a previous study conducted by Chakraborty et al. [30] that revealed that *M. oleifera* leaf extract is effective against *S. typhimurium*. Likewise, Adeyemi et al. [31] found that *Salmonella* was absent in all treated fish with *M*. *oleifera*. Additionally, Bukar et al. [32] declared the bioactivity of *M*. *oleifera* ethanol extract on pathogenic bacteria, for example, *S. typhimurium*, *E*. *coli*, and *Enterobacter*. Consequently, *M*. *oleifera* extracts could be used as promising food additives that possess antimicrobial properties against both spoilage and pathogenic microorganisms, which can enhance the storage life of meat and meat products and increase the profitability of the meat industry.

#### 3.2.3. Antimicrobial Effect of MLE against *S. aureus*

The current study revealed no significant difference (*p* > 0.05) in the counts of inoculated *S. aureus* amongst control and treated beef meatball samples with 0.5%, 1%, and 2% MLE on Day 0 and Day 3 (Table 1). However, by Day 6, the counts of inoculated *S*. *aureus* in treated beef meatballs with 0.5, 1%, and 2% MLE were significantly (*p* < 0.05) decreased by 1.05, 1.22, and 1.62 log_10_ CFU/g, respectively, compared with the control sample (5.48, 5.31, and 4.91 respectively versus. 6.53 log_10_ CFU/g) (Table 1). Beef meatballs treated with 0.5%, 1%, and 2% MLE on Day 9 of storage displayed a significant (*p* < 0.05) decline in the counts of inoculated *S. aureus*, compared to the control sample, by 1.97, 2.41, and 2.94 log_10_ CFU/g, respectively. Similarly, on Day 12 of storage, treated samples with 0.5%, 1%, and 2% MLE had a significant (*p* < 0.01) decrease in *S. aureus* counts, by 2.52, 3.14, 3.77 log_10_ CFU/g, respectively, compared with the control sample (Table 1). On Day 15 of storage, beef meatballs treated with 0.5%, 1%, and 2% MLE displayed a significant (*p* < 0.01) decline in inoculated *S. aureus* counts, compared with the control sample, by 3.12, 4.23, and 4.53 log_10_ CFU/g, respectively (Table 1). At the end of the storage period (Day 18), control meatballs showed a high count of inoculated *S. aureus* of 7.51 log_10_ CFU/g, however, treated beef meatballs with 0.5%, 1%, and 2% MLE revealed a significant (*p* < 0.01) decrease in *S. aureus* counts, by 3.82, 4.97, and 5.40 log_10_ CFU/g, respectively, when compared with the control (Table 1). 

A previous study conducted by Moyo et al. [28] showed that MLE had antimicrobial properties by inhibiting the growth of *S*. *aureus* strains isolated from food and animal intestines. Jayawardana et al. [23] mentioned that *M*. *oleifera* leaves contain a chemical substance named pterygospermin that readily separates into two molecules of benzyl isothiocyanate, which is known to have antimicrobial properties. The same authors found that *S*. *aureus* was less than 10^2^ CFU per gram in chicken sausages treated with 0.25%, 0.5%, 0.75%, and 1% *M. oleifera* leaves. Furthermore, Elhadi et al. [29] found that *S*. *aureus* counts were lower in refrigerated chicken patties treated with 100 g/kg MLP, than in chicken patties treated with 50 g/kg MLP and control patties (without treatment), throughout the storage periods (12 days). The same authors revealed that the antimicrobial properties of *M*. *oleifera* could be attributed to its contents of phytochemical compounds, for instance, polyphenols, flavonoids, other proteins, and peptides.

### 3.3. Sensory Evaluation

*Moringa oleifera* is a nutraceutical element rich in essential amino acids, minerals, and vitamins. *M. oleifera* leaves are considered a miracle food that could give enormous nutrition to people suffering from malnutrition and may be considered a protein and calcium supplement [33]. Moringa flavor intensity, characteristic flavor for beef meatballs, tenderness, juiciness, and the overall acceptability of treated and control beef meatball samples are shown in Table 2.

*Moringa* flavor intensity was significantly (*p* < 0.01) detected in treated beef meatballs with 0.5%, 1%, and 2% MLE throughout the storage periods (Table 2). *M. oleifera* leaves are rich in polyphenols, carotenoids, flavonoids, and other bioactive compounds, which give a favorable taste and aroma to foods [11]. Moreover, there was no significant difference detected in the characteristic flavor of beef meatballs, tenderness, juiciness, and overall acceptability between treated and control beef meatball samples; however, a slight improvement in both tenderness and juiciness was observed in treated meatball samples in comparison to the control (Table 2). On the other hand, Rahman et al. [26] found a significant (*p* < 0.05) increase in the color, flavor, tenderness, juiciness, and overall acceptability of goat meat nuggets treated with 0.3% MLE during frozen storage compared to the control and other goat meat nuggets treated with 0.1% BHA. Furthermore, Evivie et al. [33] revealed that soy meatballs treated with 1%, 2%, 3%, and 4% *M. oleifera* leaves powder, as well as control samples, were generally accepted by panelists, although the addition of 2% or more of *M. oleifera* leaf powder to the meatballs decreased the panelists’ acceptance, while 1% *M. oleifera* leaf powder had the same panelist acceptance rate as the control samples. Additionally, the addition of *M. oleifera* leaf powder to chicken patties up to concentrations of 50 g/kg did not affect the overall acceptability and other sensory parameters of chicken patties [29]. 

The shelf-life of meat products is greatly affected by microbiological, chemical, and sensory evaluations. In the present study, a high count of 10^7^ CFU/g for APC and 10^4^ CFU/g for EBC or more is usually associated with deteriorative changes that affect the sensory attributes and determine the product shelf life. Due to deteriorative changes noticed by changes in odor and color, sensory evaluation was not conducted from Day 9 and thereafter for control samples, from Day 12 and thereafter for beef meatballs treated with 0.5% MLE, and from Day 15 and afterward for beef meatballs treated with 1% MLE (Table 2). *Moringa* leaf extracts subjected to boiling for at least 15 min had higher antioxidant activity than raw leaves because boiling induced a significant increase in the proximate composition of leaves extract [34]. Moreover, MLE significantly increased the cooking yield and moisture and fat retention of treated mutton patties. Thus, the incorporation of MLE in meat products could help in the formation of a strong structure of meat products [27]. Therefore, MLE could be used as a valuable, natural, and safe preservative to improve the nutritional value, organoleptic properties, and shelf-stability of meat products. Similar to our findings, the *M. oleifera* flower increased the odor score, lipid stability, and shelf-life stability of chicken nuggets during cold storage for 20 days [35]. 

## 4. Conclusions

The present study indicates that adding *Moringa oleifera* leaves extract to ground beef has potent antimicrobial activity against food borne pathogens. Generally, *M. oleifera* leaves aqueous extracts 2% was the most effective treatment for controlling the growth of foodborne pathogens artificially inoculated to beef meatballs, including *E. coli* O157:H7, *S. enterica* serovar Typhimurium, and *S. aureus*, followed by 1% and 0.5% aqueous extracts of *M. oleifera* leaves. Fortunately, *M. oleifera* leaves aqueous extract did not affect the overall acceptability and the other sensory parameters of ground beef. Consequently, *M*. *oleifera* leaves extracts can be used as natural, safe, and cheap food additives to control the growth of foodborne pathogens and increase the shelf life of meat products. More research about the antimicrobial effect of *M*. *oleifera* leaves extract is required.

## Figures and Tables

**Figure 1 foods-12-00766-f001:**
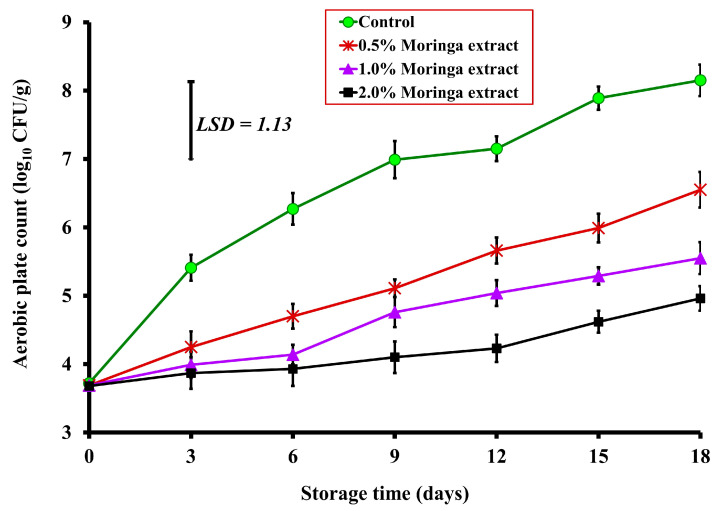
Effect of *Moringa oleifera* leaves extract supplemented to ground beef on the aerobic plate count (APC) (log_10_ CFU/g) during cold storage at 4 °C for 18 days. The displayed values represented the mean of triplicate measurements ± standard error (SE).

**Figure 2 foods-12-00766-f002:**
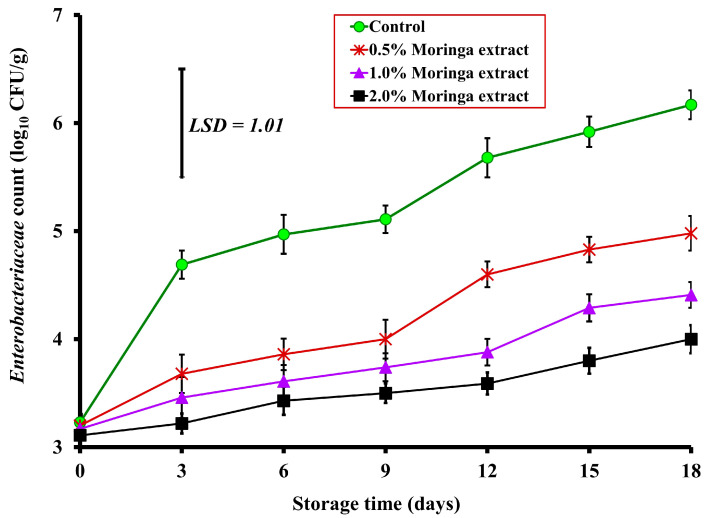
Effect of *Moringa oleifera* leaves extract supplemented to ground beef on the Enterobacteriaceae counts (log_10_ CFU/g) during cold storage at 4 °C for 18 days. The displayed values represented the mean of triplicate measurements ± standard error (SE).

**Table 1 foods-12-00766-t001:** Effect of *Moringa oleifera* leaves extract supplemented to ground beef on artificially inoculated *E. coli* O157:H7, *S. enterica* serovar Typhimurium, and *S. aureus* counts (log_10_ CFU/g) during cold storage at 4 °C for 18 days.

Microbial Category	Storage Day	Control	0.5% *Moringa* Extract	1% *Moringa* Extract	2% *Moringa* Extract
*E. coli* O157:H7 counts	0	6.69 ^a^ ± 0.17	6.67 ^a^ ± 0.16	6.63 ^a^ ± 0.16	6.59 ^a^ ± 0.17
	3	7.15 ^a^ ± 0.16	6.41 ^a,b^ ± 0.16	6.11 ^b^ ± 0.14	5.62 ^b^ ± 0.12
	6	7.34 ^a^ ± 0.15	6.00 ^b^ ± 0.16	5.66 ^b,c^ ± 0.12	4.98 ^c^ ± 0.09
	9	7.92 ^a^ ± 0.15	5.89 ^b^ ± 0.17	5.12 ^b,c^ ± 0.05	4.22 ^c^ ± 0.11
	12	8.09 ^a^ ± 0.14	5.38 ^b^ ± 0.09	4.70 ^b^ ± 0.04	3.59 ^c^ ± 0.04
	15	8.26 ^a^ ± 0.12	5.00 ^b^ ± 0.07	3.99 ^c^ ± 0.08	2.87 ^d^ ± 0.05
	18	8.54 ^a^ ± 0.10	4.84 ^b^ ± 0.07	3.61 ^c^ ± 0.05	2.00 ^d^ ± 0.08
*Salmonella enterica* serovar Typhimurium counts	0	6.47 ^a^ ± 0.16	6.45 ^a^ ± 0.15	6.43 ^a^ ± 0.16	6.41 ^a^ ± 0.16
	3	6.89 ^a^ ± 0.15	6.15 ^a^ ± 0.15	6.00 ^a^ ± 0.15	5.92 ^a^ ± 0.14
	6	7.01 ^a^ ± 0.15	6.00 ^b^ ± 0.15	5.81 ^b,c^ ± 0.15	4.99 ^c^ ± 0.12
	9	7.26 ^a^ ± 0.14	5.87 ^b^ ± 0.11	5.41 ^b,c^ ± 0.12	4.53 ^c^ ± 0.11
	12	7.58 ^a^ ± 0.13	5.51 ^b^ ± 0.09	5.12 ^b^ ± 0.09	4.00 ^c^ ± 0.09
	15	7.90 ^a^ ± 0.10	5.23 ^b^ ± 0.06	4.11 ^c^ ± 0.07	3.62 ^c^ ± 0.07
	18	8.05 ^a^ ± 0.12	4.95 ^b^ ± 0.04	3.74 ^c^ ± 0.05	2.70 ^d^ ± 0.03
*Staphylococcus aureus* counts	0	6.01 ^a^ ± 0.16	6.00 ^a^ ± 0.15	5.96 ^a^ ± 0.15	5.90 ^a^ ± 0.15
	3	6.17 ^a^ ± 0.14	5.82 ^a^ ± 0.14	5.61 ^a^ ± 0.13	5.54 ^a^ ± 0.13
	6	6.53 ^a^ ± 0.13	5.48 ^b^ ± 0.14	5.31 ^b^ ± 0.15	4.91 ^b^ ± 0.12
	9	6.94 ^a^ ± 0.12	4.97 ^b^ ± 0.12	4.53 ^b^ ± 0.12	4.00 ^b^ ± 0.09
	12	7.11 ^a^ ± 0.13	4.59 ^b^ ± 0.09	3.97 ^b,c^ ± 0.07	3.34 ^c^ ± 0.11
	15	7.23 ^a^ ± 0.12	4.11 ^b^ ± 0.11	3.00 ^c^ ± 0.09	2.70 ^c^ ± 0.08
	18	7.51 ^a^ ± 0.11	3.69 ^b^ ± 0.07	2.54 ^c^ ± 0.11	2.11 ^c^ ± 0.04

The displayed values represented the mean of triplicate measurements ± standard error (SE). Mean values with a different superscript letter in the same row are significantly different (*p* < 0.05 and *p* < 0.01).

**Table 2 foods-12-00766-t002:** * Mean values of the sensory characteristic score of beef meatballs treated with different concentrations of *Moringa oleifera* leaf extract during cold storage at 4 °C for 18 days.

Storage Day	Sensory Characteristics	Control Beef Meatball	*Moringa oleifera* Leaf Extract-Treated Ground Beef		
			0.5%	1%	2%
Day 0	*Moringa* flavor *^x^*	1.98 ^a^	5.27 ^b^	5.74 ^b,c^	6.39 ^c^
	Characteristic beef meatballs flavor *^x^*	6.89 ^a^	6.64 ^a^	6.51 ^a^	6.40 ^a^
	Tenderness *^x^*	6.63 ^a^	7.25 ^a^	7.08 ^a^	7.17 ^a^
	Juiciness *^x^*	6.30 ^a^	6.46 ^a^	6.67 ^a^	6.79 ^a^
	Overall acceptability *^y^*	7.91 ^a^	7.82 ^a^	7.74 ^a^	7.70 ^a^
Day 3	*Moringa* flavor *^x^*	2.05 ^a^	5.19 ^b^	5.63 ^b,c^	6.12 ^c^
	Characteristic beef meatballs flavor *^x^*	6.72 ^a^	6.37 ^a^	6.31 ^a^	6.24 ^a^
	Tenderness *^x^*	6.69 ^a^	7.29 ^a^	7.04 ^a^	7.31 ^a^
	Juiciness *^x^*	6.23 ^a^	6.38 ^a^	6.75 ^a^	6.68 ^a^
	Overall acceptability *^y^*	7. 79 ^a^	7.65 ^a^	7.71 ^a^	7.78 ^a^
Day 6	*Moringa* flavor *^x^*	2.03 ^a^	5.11 ^b^	5.56 ^b,c^	6.03 ^c^
	Characteristic beef meatballs flavor *^x^*	6.58 ^a^	6.32 ^a^	6.20 ^a^	6.13 ^a^
	Tenderness *^x^*	6.61 ^a^	7.18 ^a^	7.09 ^a^	7.24 ^a^
	Juiciness *^x^*	6.15 ^a^	6.35 ^a^	6.64 ^a^	6.67 ^a^
	Overall acceptability *^y^*	7.53 ^a^	7.62 ^a^	7.68 ^a^	7.70 ^a^
Day 9	*Moringa* flavor *^x^*	ND	4.83 ^b^	5.26 ^b^	5.81 ^c^
	Characteristic beef meatballs flavor *^x^*		6.27 ^a^	6.11 ^a^	6.06 ^a^
	Tenderness *^x^*		7.21 ^a^	7.13 ^a^	7.28 ^a^
	Juiciness *^x^*		6.43 ^a^	6.69 ^a^	6.81 ^a^
	Overall acceptability *^y^*		7.49 ^a^	7.58 ^a^	7.66 ^a^
Day 12	*Moringa* flavor *^x^*	ND	ND	4.72 ^b^	5.10 ^b^
	Characteristic beef meatballs flavor *^x^*			6.05 ^a^	6.01 ^a^
	Tenderness *^x^*			7.02 ^a^	7.19 ^a^
	Juiciness *^x^*			6.68 ^a^	6.77 ^a^
	Overall acceptability *^y^*			7.56 ^a^	7.62 ^a^
Day 15	*Moringa* flavor *^x^*	ND	ND	ND	4.84
	Characteristic beef meatballs flavor *^x^*				5.89
	Tenderness *^x^*				7.15
	Juiciness *^x^*				6.73
	Overall acceptability *^y^*				7.58
Day 18	*Moringa* flavor *^x^*	ND	ND	ND	4.71
	Characteristic beef meatballs flavor *^x^*				5.80
	Tenderness *^x^*				7.13
	Juiciness *^x^*				6.69
	Overall acceptability *^y^*				7.55

* Mean values with a different superscript letter in the same row are significantly different (*p* < 0.05); ND: not detected due to deteriorative changes; *^x^* an eight-point hedonic scoring scale from 1 (an extremely unacceptable sample) to 8 (an extremely acceptable sample) was used to judge each of *Moringa* flavor intensity, characteristic flavor for beef meatball, tenderness, and juiciness; *^y^* a nine-point hedonic scale from 1 (an extremely disliked sample) to 9 (an extremely liked sample) was used to determine the overall acceptability.

## Data Availability

All data generated or analyzed during this study are included in the submitted version of the manuscript.

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
