# Peer review of "Antimicrobial Effect of Moringa oleifera Leaves Extract on Foodborne Pathogens in Ground Beef"

_foods, 2023, doi:10.3390/foods12040766_

Round 1

Reviewer 1 Report

MDPI - Foods: Manuscript: foods-2182084

Antimicrobial effect of Moringa oleifera leaves extract on foodborne pathogens in ground beef

By: R. Abdallah et al.

General comments:

This manuscript reports findings on the antimicrobial effect of Moringa oleifera leaves extract (MLE) at different concentration on ground beef during refrigerated storage. The idea of using natural herbal based preservatives for extending the shelf life of foods itself is interesting to follow. The authors have used the aqueous extract of Moringa oleifera leaves for preservation of minced beef. The have shown that MLE provide antimicrobial properties against several spoilage bacteria. Meanwhile, MLE showed no adverse effect on the overall acceptability of meat and somehow induced a slight improvement in meat tenderness. It has been claimed that MLE can be used as a healthy, natural, and safe preservative to increase meat products' safety, quality, and shelf stability during cold storage.

In general, this study covers a number of useful characterization techniques to evaluate the research hypothesis. In my opinion the topic and research subject are interesting and has novelty. However, I would suggest flowing points to be considered by authors to improves the quality of this manuscript.

Abstract:

Abstract has been written well; however following points can be considered to improve it:

·         hypothesis behind doing this study (after a short introduction)

·         future prospective (implication of the results for industrialization natural preservatives in meat industry including MLE).

·         I would suggest authors to include antimicrobial results prior than sensory results.

Introduction:

Introduction is too short. Please give a brief survey of literature about similar reports on incorporating natural preservatives for meat products from agricultural wastes and then explain the novelty of your research compered to those literature. Also, provide the “research hypothesis” at the end of your introduction. English grammar in some parts needs to be improved.

Materials & Methods:

I would prefer an order of methods: Physicochemical, Microbial, Sensorial. This order should be the same in Results and Discussion.

Section 2.3, lines 126-128: Please provide more details about your sensory test including evaluation sheet.

Results and Discussion:

Similar to M & M, Physicochemical, Microbial, Sensorial order should be followed in Results and Discussion.

Chemical characterization of Moringa extract as well as physicochemical characteristics of your meat sample should come first in the Results section.

Please provide discussion about chemical and thermal instability of Moringa extract during meat processing (cooking, canning, roasting, etc.)

Conclusions

Good.

Author Response

Reviewer #1:

This manuscript reports findings on the antimicrobial effect of Moringa oleifera leaves extract (MLE) at different concentration on ground beef during refrigerated storage. The idea of using natural herbal based preservatives for extending the shelf life of foods itself is interesting to follow. The authors have used the aqueous extract of Moringa oleifera leaves for preservation of minced beef. The have shown that MLE provide antimicrobial properties against several spoilage bacteria. Meanwhile, MLE showed no adverse effect on the overall acceptability of meat and somehow induced a slight improvement in meat tenderness. It has been claimed that MLE can be used as a healthy, natural, and safe preservative to increase meat products' safety, quality, and shelf stability during cold storage.

In general, this study covers a number of useful characterization techniques to evaluate the research hypothesis. In my opinion the topic and research subject are interesting and has novelty. However, I would suggest flowing points to be considered by authors to improves the quality of this manuscript.

Dear reviewer,

Thank you so much for your review, kind comments, and valuable suggestions. We have modified the text according to them.

Abstract:

Abstract has been written well; however following points can be considered to improve it:

- hypothesis behind doing this study (after a short introduction).

It is added in the revised manuscript.

- future prospective (implication of the results for industrialization natural preservatives in meat industry including MLE).

Added in the revised manuscript.

- I would suggest authors to include antimicrobial results prior than sensory results.

It is rearranged in the revised manuscript.

Introduction:

Introduction is too short. Please give a brief survey of literature about similar reports on incorporating natural preservatives for meat products from agricultural wastes and then explain the novelty of your research compared to those literature. Also, provide the “research hypothesis” at the end of your introduction. English grammar in some parts needs to be improved.

Literature about similar reports incorporating natural preservatives for meat products is added. The research hypothesis is clearly mentioned at the end of the revised introduction.

English grammar is improved in the revised manuscript.

Materials & Methods:

I would prefer an order of methods: Physicochemical, Microbial, Sensorial. This order should be the same in Results and Discussion.

The order is rearranged according to your suggestion both in the “materials and methods” and in the “results and discussion” sections.

Section 2.3, lines 126-128: Please provide more details about your sensory test including evaluation sheet.

Additional details are provided in the revised manuscript.

Results and Discussion:

Similar to M & M, Physicochemical, Microbial, Sensorial order should be followed in Results and Discussion.

The order is rearranged according to your suggestion in the “Results and discussion” sections.

Chemical characterization of Moringa extract as well as physicochemical characteristics of your meat sample should come first in the Results section.

The order is rearranged according to your suggestion.

Please provide discussion about chemical and thermal instability of Moringa extract during meat processing (cooking, canning, roasting, etc.).

It is added in the revised manuscript in the “results and discussion” sections.

Conclusions: Good.

Thank you again for your time and suggestions!

Reviewer 2 Report

The paper “Antimicrobial effect of Moringa oleifera leaves extract on foodborne pathogens in ground beef” evaluate the antimicrobial effects of three different concentrations of Moringa oleifera leaves extract on beef meatballs during shelf life. The antimicrobial effect was evaluated on three different pathogens and on Aerobic colony count and Enterobacteriaceae mean counts. Moreover, sensory evaluation was conducted.

The manuscript contains interesting results. However, it must be deeply improved.

English is poor and the manuscript should be revised by a native speaker. Just some examples:

Line 36: change “their” with “its” as it is referred to the word “meat”.

Line 44: change in “APC is considered a gold standard…”.

Line 190: “M. oleifera leaves are considered…”

And so on throughout the manuscript.

Avoid generic sentences like lines 61-64: “Additionally, consumers 61 and producers are becoming more open to using only meat products containing safe and natural additives, because the accumulation of synthetic chemical preservatives in human 63 tissues could have a detrimental effect on health” or lines 189-192 “M. oleifera leaves consider a miracle food that could give enormous nutrition to people suffering from malnutrition and may be considered a protein and calcium supplement”.

Line 36: what do the authors mean with “high pH”? Beef meat pH is usually comprised between 5.5 and 5.8.

Line 41: change “suppress” with “reduce”.

Avoid using generic terms like “higher counts” or “higher levels” (line 45 and 48). Always refer to precise values of microbial counts (or at least a range of values”.

Also avoid expressions like “mixed well” (line 171) that denotes a very poor scientific approach.

Aerobic plate count and Enterobacteriaceae cannot be considered as spoilage indicators but as indicators of the process hygiene. Also correct the sentence at line 399 in the conclusions section.

Line 108-109: please clarify if sterile instruments were used for sampling, do not refer to generic statements as “under sanitary conditions” or “sharp knife”.

Line 143 and following. Give details about the strains used for inoculum (reference strains or wild strains).

It would be interesting to evaluate if significant changes occurred not only between treatments but also, within the same treatment, during time.

In the results section avoid sentence like (line 238-239 and 282) “high reduction rate” but refer to “significant reduction rate”.

Based on the results of the sensory evaluation (Line 221-225) and of the microbial counts, could the authors determine the shelf life of the meatballs?  

Lines 270 and following: Salmonella, Shigella etc. belong to the order Enterobacterales.

Author Response

Reviewer #2:

The paper “Antimicrobial effect of Moringa oleifera leaves extract on foodborne pathogens in ground beef” evaluate the antimicrobial effects of three different concentrations of Moringa oleifera leaves extract on beef meatballs during shelf life. The antimicrobial effect was evaluated on three different pathogens and on Aerobic colony count and Enterobacteriaceae mean counts. Moreover, sensory evaluation was conducted.

The manuscript contains interesting results. However, it must be deeply improved.

Dear reviewer,

Thank you for your efforts and time for the revision of our manuscript!

English is poor and the manuscript should be revised by a native speaker. Just some examples:

The authors acknowledge the fact that the English content of the manuscript need some improvements. According to the reviewer suggestion, during the manuscript revision, a native English speaker colleague helped the research team to improve the English content of the manuscript. In addition, the authors consider that the improvement of the English language is solvable during the final “Pending English” status before publication, when the paper will be edited and finalized by the MDPI team.

Thank you for your understanding and consideration!

Line 36: change “their” with “its” as it is referred to the word “meat”. Changed

Line 44: change in “APC is considered a gold standard…”. Changed

Line 190: “M. oleifera leaves are considered…” Changed

And so on throughout the manuscript.

Avoid generic sentences like lines 61-64: “Additionally, consumers 61 and producers are becoming more open to using only meat products containing safe and natural additives, because the accumulation of synthetic chemical preservatives in human 63 tissues could have a detrimental effect on health” or lines 189-192 “M. oleifera leaves consider a miracle food that could give enormous nutrition to people suffering from malnutrition and may be considered a protein and calcium supplement”.

Avoided in the revised manuscript.

Line 36: what do the authors mean with “high pH”? Beef meat pH is usually comprised between 5.5 and 5.8.

The word ”high” was changed to suitable.

Line 41: change “suppress” with “reduce”.

Changed

Avoid using generic terms like “higher counts” or “higher levels” (line 45 and 48). Always refer to precise values of microbial counts (or at least a range of values”.

Revised and corrected.

Also avoid expressions like “mixed well” (line 171) that denotes a very poor scientific approach.

Revised and corrected.

Aerobic plate count and Enterobacteriaceae cannot be considered as spoilage indicators but as indicators of the process hygiene. Also correct the sentence at line 399 in the conclusions section.

Revised and corrected.

Line 108-109: please clarify if sterile instruments were used for sampling, do not refer to generic statements as “under sanitary conditions” or “sharp knife”.

Revised and corrected.

Line 143 and following. Give details about the strains used for inoculum (reference strains or wild strains).

Details are added in the revised manuscript. The strains used were wild-type previously isolated from meat products in our laboratory.

It would be interesting to evaluate if significant changes occurred not only between treatments but also, within the same treatment, during time.

In the revised manuscript, a new paragraph concerning the time effect for each treatment is added in 3.1.1. for APC and also in 3.1.2. for EBC.

In the results section avoid sentence like (line 238-239 and 282) “high reduction rate” but refer to “significant reduction rate”.

Revised and corrected.

Based on the results of the sensory evaluation (Line 221-225) and of the microbial counts, could the authors determine the shelf life of the meatballs?

A clause concerning this issue is added in the discussion section under 3.3.

Lines 270 and following: Salmonella, Shigella etc. belong to the order Enterobacterales.

Such new taxonomy is mentioned in the revised manuscript.

Reviewer 3 Report

This manuscript presents a study about the antimicrobial properties of Moringa oleifera extracts and its usefullness for preserving raw meat. The aim of the work is relevant, as the use of natural antimicrobial compounds is a promising way to combine the need for extended shelf-life produts with the requierementes for less chemical additives in processed food.

Experimental work is well designed, results are clearly presented and conclusions are coherent with results. 

My only concern is about the originality of the work. Previous similar studies have been already published and authors themselves use these previous works to compare and discuss their results. Thus, it should be important than authors remark in the manuscript what addition to scientific knowledge makes this work in comparison with previous literature on this subject.

Author Response

Reviewer #3:

This manuscript presents a study about the antimicrobial properties of Moringa oleifera extracts and its usefullness for preserving raw meat. The aim of the work is relevant, as the use of natural antimicrobial compounds is a promising way to combine the need for extended shelf-life produts with the requierementes for less chemical additives in processed food.

Experimental work is well designed, results are clearly presented and conclusions are coherent with results.

Thank you so much for your review, kind comments, and valuable suggestions. We have modified the text according to them.

My only concern is about the originality of the work. Previous similar studies have been already published and authors themselves use these previous works to compare and discuss their results. Thus, it should be important than authors remark in the manuscript what addition to scientific knowledge makes this work in comparison with previous literature on this subject.

Studies concerning the antimicrobial effect of MLE against foodborne pathogens are scarce and mostly focused on its in vitro effect against some foodborne pathogens. Only two in vivo studies were carried out on chicken products against E. coli and S. aureus. This study, is therefore, very important to elucidate the antibacterial effect of MLE in meat products. A sentence declaring the importance and the originality of this work to scientific knowledge was added.

Thank you again for your time and efforts!

Reviewer 4 Report

The purpose of this manuscript was to investigate the effect of the aqueous extract of Moringa oleifera leaves on the indigenous microbiota (determined as aerobic plate counts and Enterobacteriaceae) and on selected food-borne pathogens. The paper provides interesting information on the potential use of the leaf extract from this tree as a novel antimicrobial compound. I have the following comments on this work:

Page 3, line 116: The authors focused their attention on the determination of aerobic plate counts (APC) and Enterobacteriaceae. Please justify why other important microorganisms such as Pseudomonas spp., B. thermosphacta were not taken into consideration. Both bacteria belong to the specific spoilage microorganisms of beef.

Page 3, line 127: In the sensory analysis undertaken by the 15 experienced panelists, the attributes assessed were the characteristic flavor of beef, tenderness and juiciness. How were the panelists trained on these attributes? How can you score e.g. the characteristics flavor of beef on a 8-point scale? Why did the authors use a 9-point scale for the assessment of the overall acceptability of the samples? In the caption of Table 1, the 1-8 hedonic scale refers to an “extremely acceptable sample” (score 8) and “extremely unacceptable sample” (score 1). How could the panelists judge what is “extremely acceptable” or “extremely unacceptable”?   

Page 3, line 138: VRBGA is normally incubated at 37°C for 24 h and not at 30°C for 48 h. In addition this medium is inoculated by the spread technique (using 1 mL of bacterial suspension) and not with the spread technique (using 0.1 mL). Please justify.

Page 7, line 142: the plural for the word “inoculum” is “inocula” and not “inoculums”. Please correct.

Line 231, paragraph 3.2.1: The authors are strongly advised to present the data as figures, not tables.

Table 2: From this table it is clear that there is a decreasing trend of the pathogens in all cases, the highest being in 2% leaf extract. However, there is remaining population of foodborne pathogens in the end of storage and this is due to the fact the authors used a rather high initial inoculum. How realistic is it to have 6 logs of Salmonella in the beginning of storage in beef? Why the authors did not choose a lower inoculum level that would be more realistic? What would be the fate of the pathogens if the initial inoculum was 2 or 3 logs instead of 6? By the way, what was the criterion used by the authors to define the storage time at 18 days?

Author Response

Reviewer #4:

The purpose of this manuscript was to investigate the effect of the aqueous extract of Moringa oleifera leaves on the indigenous microbiota (determined as aerobic plate counts and Enterobacteriaceae) and on selected food-borne pathogens. The paper provides interesting information on the potential use of the leaf extract from this tree as a novel antimicrobial compound. I have the following comments on this work:

Dear reviewer, Thank you so much for your review, kind comments, and valuable suggestions. We have modified the text according to them.

Page 3, line 116: The authors focused their attention on the determination of aerobic plate counts (APC) and Enterobacteriaceae. Please justify why other important microorganisms such as Pseudomonas spp., B. thermosphacta were not taken into consideration. Both bacteria belong to the specific spoilage microorganisms of beef.

The title and the main purpose of this manuscript focus on the effect of Moringa leaves extract on three of the most important foodborne pathogens. The APC and Enterobacteriaceae count along with the sensory attributes were carried out to give an idea about the influence of Moringa extract on the shelf life of ground beef during cold storage.

Page 3, line 127: In the sensory analysis undertaken by the 15 experienced panelists, the attributes assessed were the characteristic flavor of beef, tenderness and juiciness. How were the panelists trained on these attributes? How can you score e.g. the characteristics flavor of beef on a 8-point scale? Why did the authors use a 9-point scale for the assessment of the overall acceptability of the samples? In the caption of Table 1, the 1-8 hedonic scale refers to an “extremely acceptable sample” (score 8) and “extremely unacceptable sample” (score 1). How could the panelists judge what is “extremely acceptable” or “extremely unacceptable”?

All panelists who participate in the sensory evaluation of the meatballs containing the different moringa extract concentrations were trained panels received the same training and understand the descriptors and the scales being used. Panelists were able to discriminate differences between the different treated groups within the product and can describe the different attributes qualitatively, and scale the strength of the attribute quantitatively. The characteristics of Moringa flavor intensity, characteristic flavor for beef meatballs, tenderness, and juiciness were performed according to Texas Tech University - Beef Sensory Evaluation Form using 8-point scale (https://www.depts.ttu.edu/meatscience/docs/TTUBeefSensoryForm.pdf), whereas the overall acceptability of beef meatballs tested was evaluated using a 9-hedonic scales where the participants indicate how much they like or dislike the sample in terms of a specific sensory property, such as appearance, flavor, taste, and texture, and can also include overall liking/acceptance. The most commonly used scale is the 9-point hedonic scale that ranges from “like extremely” to “dislike extremely”.

Page 3, line 138: VRBGA is normally incubated at 37°C for 24 h and not at 30°C for 48 h. In addition, this medium is inoculated by the spread technique (using 1 mL of bacterial suspension) and not with the spread technique (using 0.1 mL). Please justify.

Thank you for your observations. It was mistakenly written as that of the enumeration of aerobic plate counts. We exactly perform the same procedures you described and it is now corrected in the revised manuscript.

Page 7, line 142: the plural for the word “inoculum” is “inocula” and not “inoculums”. Please correct.

Corrected

Line 231, paragraph 3.2.1: The authors are strongly advised to present the data as figures, not tables.

Table 2 (in the original submission) is replaced in the revised manuscript by Figure 1 for APC and Figure 2 for EBC.

Table 2: From this table it is clear that there is a decreasing trend of the pathogens in all cases, the highest being in 2% leaf extract. However, there is remaining population of foodborne pathogens in the end of storage and this is due to the fact the authors used a rather high initial inoculum. How realistic is it to have 6 logs of Salmonella in the beginning of storage in beef? Why the authors did not choose a lower inoculum level that would be more realistic? What would be the fate of the pathogens if the initial inoculum was 2 or 3 logs instead of 6? By the way, what was the criterion used by the authors to define the storage time at 18 days?

We start the experiment using inoculum of high count in order to recognize the reduction by logs among the meatball’s groups treated with different moringa concentrations used. If we started with 2 or 3 logs, a reduction by logs will not be recognized since log 1 is almost uncountable. The experiment was ended at the day 18 since the control, 0.5, and 1.0% moringa leaves extract-treated ground beef samples were deteriorated based on the sensorial and microbial evaluations.

Thank you so much!